# Placental Transfer Immunity to the Newborns in a Twin Pregnant Women Vaccinated with Heterologous CoronaVac-ChAdOx1

**DOI:** 10.3390/vaccines11010116

**Published:** 2023-01-03

**Authors:** Saipin Pongsatha, Kriangkrai Chawansuntati, Supachai Sakkhachornphop, Theera Tongsong

**Affiliations:** 1Department of Obstetrics and Gynecology, Faculty of Medicine, Chiang Mai University, Chiang Mai 50200, Thailand; 2Research Institute for Health Sciences, Chiang Mai University, Chiang Mai 50200, Thailand

**Keywords:** COVID-19, heterologous CoronaVac-ChAdOx1, placental transfer, spike protein, twin pregnancy, vaccine

## Abstract

Pregnant women who receive the COVID-19 vaccine develop anti-SARS-CoV-2 antibodies, which can be transferred to the fetus. However, the effectiveness of placental transfer has not been evaluated in twin pregnancy, especially in cases vaccinated with heterologous CoronaVac (Sinovac)—ChAdOx1 (Oxford-AstraZeneca) regimen, which was commonly used in many countries. Case: A 34-year-old Thai woman with a twin pregnancy attended our antenatal care clinic at 21 + 2 weeks of gestation and requested COVID-19 vaccination. Her medical history and physical examination were unremarkable. She had not received COVID-19 vaccination before. Ultrasound screening for fetal anomaly revealed a dichorion diamnion twin pregnancy. Both twins showed no structural anomaly. She received the CoronaVac vaccine at 21 + 2 weeks of gestation without serious side effects and the ChAdOx1 vaccine at 24 + 2 weeks of gestation. Cesarean delivery was performed at 36 + 5 weeks of gestation, giving birth to the two healthy babies. The levels of anti-spike protein IgG levels (BAU/mL) in maternal blood just before delivery and umbilical cord blood of the two newborns were 313.349, 678.219, and 874.853, respectively. The levels of % inhibition (wild-type and delta) in the two newborns were also higher than those in the mother. In conclusion, heterologous CoronaVac-ChAdOx1-S vaccination in a twin pregnancy could effectively provide protective immunity to both twin newborns. The antibody levels in both were approximately two times higher than those in the mothers. This case report may serve as a reference in counseling couples with a twin pregnancy, while the studies on placental transfer of vaccine-derived antibodies in twin pregnancy are currently not available, especially in countries experiencing a vaccine shortage or unavailability of mRNA vaccines.

## 1. Introduction

Pregnant women have a higher risk of severe COVID-19 infection [1,2,3,4,5]. The risk may be associated with adaptive changes occurring during pregnancy, such as the reduction in the residual lung capacity, the decrease in viral immune responses, and the increased risk for thrombotic events. The risk may be expected to be even higher among twin pregnancies. Accordingly, COVID-19 vaccination of pregnant women is very important to reduce maternal and perinatal morbidity/mortality [6]. Additionally, IgG antibodies derived from vaccination can be transferred across the placenta to the fetuses and newborns [7,8], as seen in natural infection. Placental transfer antibody seems to be safe for fetuses and is expected to have a protective effect on newborns [9], though the effectiveness of such protection has not been thoroughly evaluated. To date, a very limited number of studies on COVID-19 vaccination with placental transfer have been published, and most are mRNA vaccines [10,11,12,13]. In low- and middle-income countries facing a vaccine shortage, especially of mRNA vaccines, and with emerging variants, the heterologous COVID-19 vaccine regimen has the potential to accelerate vaccine rollout. Two-dose vaccination administered in a short time could rapidly increase protective immunity within the population in the middle of the COVID-19 pandemic with emerging variants. Heterologous CoronaVac (Sinovac) followed by ChAdOx1-S (Oxford–AstraZeneca) can induce SARS-CoV-2 RBD-specific antibodies and neutralizing activities against Wuhan type and variants similar to the licensed two-dose ChAdOx1-S [14]. In Thailand, heterologous CoronaVac-ChAdOx1 is one of the preferred regimens recommended by the National Vaccine Committee. This prime-boost vaccine regimen has been proven to be safe and offer higher immunogenicity with a shorter duration to peak immunogenicity compared to the homologous CoronaVac schedule [14,15].

The objective of this report is to illustrate the immunologic response in a twin pregnancy to the heterologous CoronaVac-ChAdOx1 vaccine, which has never been evaluated in terms of placental transfer to protect the newborns. This report may be helpful, as a piece of evidence, in counseling couples or women with twin pregnancy, in terms of its safety and effectiveness in maternal protection and placental transfer of immunity to both twins, while the studies on COVID-19 vaccine in twin pregnancies are currently not yet available, especially in the geographical areas where heterologous CoronaVac-ChAdOx1 vaccine is commonly used.

## 2. Case Presentation

A 34-year-old pregnant Thai woman, gravida 2, with previous spontaneous abortion at 10 weeks of gestation, attended our antenatal care clinic at 21 + 2 weeks of gestation. She had no familial diseases or any underlying medical disease. Physical examination revealed unremarkable findings. Basic laboratory results of standard antenatal care were all within normal limits. She had not received COVID-19 vaccination before. Ultrasound screening for fetal anomaly revealed a dichorion diamnion twin pregnancy with normal amniotic fluid volume. Both fetuses had no structural anomaly. On counseling about vaccination against COVID-19 infection, the couple chose to be vaccinated with heterologous CoronaVac-ChAdOx1. After providing written informed consent, the woman received CoronaVac (Sinovac) vaccine at 21 + 2 weeks of gestation without serious side effects and the ChAdOx1 (Oxford-AstraZeneca) vaccine at 24 + 2 weeks of gestation. She developed gestational diabetes (GDM) at 28 weeks of gestation with successful diet control and also developed preeclampsia without severe features at 36 weeks of gestation. Cesarean delivery was performed due to twin pregnancy with the non-vertex presentation of one twin with spontaneous labor at 36 + 5 weeks of gestation, 108 and 87 days after receiving the first and second dose of vaccination, respectively, giving birth to the two healthy babies, weighing 2825 and 2670 g, Apgar scores of 9 and 9 at 5 min. COVID-test (RT-PCR) in the mother during 48–72 h before delivery was negative. Postnatal examination of the placenta indicated dichorionic diamniotic type. Maternal blood and umbilical cord blood of the two newborns were collected before delivery and shortly after birth, respectively, to determine the maternal and neonatal antibody levels.

*Laboratory assays:* Anti-spike antibody levels in maternal serum and umbilical cord blood were determined as standard protocol; by validated in-house indirect ELISA. Briefly, 96-well MaxiSorp™ ELISA plates (Thermo Scientific, Roskilde, Denmark) were coated with recombinant SARS-CoV-2 Spike protein (GenScript, Piscataway, NJ, USA) overnight at 4 °C. After washing with phosphate buffer saline (PBS) containing 0.05% Tween-20 (Calbiochem, Gibbstown, NJ, USA) (PBS-T) and blocking with 2% skimmed milk in PBS, diluted serum samples and two-fold serially diluted positive controls were incubated for 1 h at 37 °C followed by washing with PBS-T. Bound antibodies were detected with horse radish peroxidase (HRP) labeled goat anti-human IgG (Invitrogen, Waltham, MA, USA) for 1 h at 37 °C. After washing with PBS-T, 3,3′,5,5′- tetramethylbenzidine (TMB) substrate (Life Technologies, Frederick, MD, USA) were added, and plates were incubated at room temperature for 20 min. The enzyme reaction was stopped using 0.2 M sulfuric acid, and absorbance was read with a CLARIOstar^®®^ microtiter plate reader (BMG Labtech, Ortenberg, Germany) at an optical density (OD) of 450 nm. Antibody levels presented as binding antibody units (BAU)/mL were determined from a standard curve of serially diluted of the WHO International Standard for anti-SARS-CoV-2 immunoglobulin (human) (code: 20/136, NIBSC, Hertfordshire, UK) which was designated an arbitrary unitage of 1000 BAU/mL. The cut-off value was assigned as ≥3 times of standard deviation of a negative test value, which was 58.5 BAU/mL.

Anti-spike antibody isotypes and IgG subclasses were also determined by replacing the secondary antibody with either mouse anti-human IgG1, mouse anti-human IgG2, mouse anti-human IgG3, mouse anti-human IgG4, mouse anti-Human IgM, or goat anti-human IgA (Invitrogen, Waltham, MA, USA) in indirect ELISA assay. The values were presented as the net OD at 450 nm, which were calculated by subtracting the OD value of wells coated with the antigen (recombinant SARS-CoV-2 Spike protein) from the OD value of antigen-free wells.

Additionally, serum-neutralizing antibodies against wild-type and delta (B.1.617.2) variants were determined using a surrogate virus neutralization test (cPass™, GenScript, Piscataway, NJ, USA) according to the manufacturer’s instruction with modifications. Briefly, serum samples and controls are pre-incubated with the receptor-binding domain (RBD)-HRP of wild-type or delta variants (catalog no. Z03614-20, GenScript, Piscataway, NJ, USA). The mixtures were then added to the capture plate pre-coated with recombinant human angiotensin-converting enzyme 2 (hACE2) protein. After washing, a TMB substrate solution was added, followed by a stop solution after the incubation period. The absorbance was read at 450 nm. The immunization profiles are presented in Table 1.

Based on the findings in Table 1, the heterologous CoronaVac-ChAdOx1-S regimen rapidly increased protective immunity in a short time in the mother and both twins. Interestingly, the levels of anti-spike protein IgG in both twins were more than two times greater than those in the mother. In addition, the percentage of inhibition against the wild-type and the delta variant was obviously greater in the newborns than those in the mother. As expected, the levels of IgM, which is typically not transferred via the placenta, were extremely low in both newborns. Unfortunately, levels of antibodies transferred to breast milk were not evaluated in this woman.

## 3. Discussion

Publications concerning the placental transfer of vaccine-induced antibodies are very limited, and all are studies on mRNA vaccine on singleton pregnancies [7,10,12,13]. To the best of our knowledge, this is the first report concerning the placental transfer of passive immunization of COVID-19 from the mother to the twin fetuses. The heterologous CoronaVac-ChAdOx1 vaccine, used in the cases presented here, was based on evidence that this regimen can result in high immunity with a shorter completion time of two doses, together with the unavailability of an mRNA vaccine. In non-pregnant subjects, the regimen can induce SARS-CoV-2 RBD-specific antibodies and neutralizing activities against wild-type and variants of concerns similar to the licensed two-dose ChAdOx1-S [14].

As already known, pregnant women are at higher risk of COVID-19 infection with more serious complications [1,2,3,4,5]. Accordingly, all pregnant women should be vaccinated against COVID-19 infection. Additionally, vaccination can probably provide a protective effect on newborns. This is one of the issues of interest. As seen in natural infection, vaccine-induced antibodies can also be transferred via the placenta to the fetuses, probably providing passive immunity to newborns. Nevertheless, data on the interval of schedules to maximize the passive immunity in newborns and type of vaccines are currently very limited. Only a few studies have been published, including BNT-162b2 (Pfizer-BioNTech) and mRNA-1273 (Moderna) vaccines [12,13,16] and a case report of inactivated virus antigen (CoronaVac^®^) [17]. The antibody levels were much higher in mothers who had received two doses of an mRNA vaccine than those who received only one dose [18,19]. Importantly, following COVID-19 vaccination, the antibodies produced are transferred to the fetus. Mithal et al. [11] showed that most pregnant women who received a COVID-19 mRNA vaccine during the third trimester had a placental transfer of IgG to the infant, with the ratio of IgG levels in cord blood to maternal blood of 1:1. However, they also showed that the transfer ratio seems to increase with latency from vaccination, from approximately 0.5 to 1.5–2 from the latency of four weeks to 10–12 weeks from first vaccination to delivery. Of interest, the findings were consistent with our report, which showed a transfer ratio of greater than two in both twins. This might be associated with a long latency period (17^+3^ weeks after 1st dose and 12^+3^ weeks after the second dose), indicating that the latency period is critical to placental transfer to the fetus.

Primary vaccine administration earlier in gestation, though a teratogenic effect is of theoretical concern in the first trimester, provides the most maternal benefit as it reduces the maternal risk of hospitalization associated with COVID-19, death from COVID-19, and COVID-19-related pregnancy complications during late pregnancy [1,6]. Although fetal and newborn antibody levels appear to be higher with primary vaccination later in pregnancy [19], this potential benefit may not outweigh the overall pregnancy (maternal, fetal, newborn) benefits of vaccination as soon as possible or account for the effects of booster doses when eligible [20]. Therefore, the best vaccination regimen (type and schedule) to provide high immunity both in the mothers and newborns remains to be elucidated. Vaccination against COVID-19 during pregnancy, if possible, should be avoided in the first trimester, and two doses are preferred. Thus, the time window for two-dose vaccination during pregnancy is relatively limited to only the second trimester and early third trimesters. Additionally, in twin pregnancy, which is commonly associated with preterm birth (before 37 weeks of gestation), it is more difficult to administer two-dose vaccination to maximize immunity in the newborns. Accordingly, a highly effective regimen with a shorter time between two-dose intervals to achieve high antibody levels in the fetuses is needed. We used the heterologous CoronaVac-ChAdOx1 vaccine in our case based on the evidence described below.

Wanlapakorn et al. [14] conducted a study comparing the effectiveness and adverse events of four vaccination regimens; homologous CV (CoronaVac) -CV and AZ (ChAdOx1) -AZ, and heterologous CV-AZ and AZ-CV. They demonstrated that receptor-binding domain (RBD)-specific antibody responses and neutralizing activities against wild-type (Wuhan) and the variants after the second dose of vaccination were much higher in the heterologous CV-AZ and homologous AZ-AZ regimens than those in the CV-CV and AZ-CV regimens; whereas adverse effects associated with vaccination were comparable in all regimens and all were minimal and well tolerated in nearly all cases. Notably, in that study, the heterologous CV-AZ regimen, which is administered four weeks apart, could induce the same levels of RBD-specific binding and neutralizing antibody as the licensed AZ-AZ vaccine, which is administered with an interval of 10–12 weeks. Furthermore, the spike-specific IgA antibodies were detected only in the CV-AZ regimen after two doses of vaccination, while the total interferon-gamma response was detected in both the CV-AZ and AZ-CV regimens after the two-dose vaccination. Therefore, if a shorter completion time of two doses is needed, such as vaccination during pregnancy, the heterologous CoronaVac-ChAdOx1 vaccine should be strongly considered. Accordingly, the heterologous CoronaVac-ChAdOx1 vaccine is attractive and has gained significant interest due to its high effectiveness, which can be achieved in a short time. A characteristic that is particularly needed during pregnancy.

This case report points out that heterologous CoronaVac-ChAdOx1 may be highly effective in providing protective immunity to newborns of twin pregnancies. The antibody levels in the newborns appear to be higher than in the mothers in both newborns. The reason why the levels in the fetuses were much higher is unclear. Hypothetically, the dilution effect of an increase in blood volume in gestation, especially twin pregnancy, may partly play a role, leading to relatively lower levels compared to those in the fetuses. Additionally, the facilitated transfer of maternally-derived antibodies through the binding of the fetal Fc receptor in the syncytiotrophoblast layer may play an essential role [21]. In addition, levels of IgG 1 and IgG 2 subtypes were higher than in the mother. Additionally, the % inhibition (wild-type and delta) was higher in both newborns. Therefore, the regimen should be considered as an alternative regimen for pregnant women, especially in low- and middle-income countries experiencing a vaccine shortage. Theoretically, passive immunity was likely effective for the protection of the twins during their neonatal life, acting similarly to the way the Tdap vaccine does. We hope this case report can encourage researchers to conduct large studies on maternal vaccination to provide neonatal immunity and improve neonatal health in the era of the COVID-19 pandemic.

It should be noted that the case presented here developed GDM and preeclampsia in late pregnancy. Nevertheless, such complications are more commonly seen in twin pregnancies. Presumably, the complications were, therefore, unlikely associated with the vaccine.

## 4. Conclusions

Heterologous CoronaVac-ChAdOx1-S vaccination in a twin pregnancy could effectively provide protective immunity to both twin newborns. The antibody levels in both were approximately two times higher than those in the mother. This case report may be served as a reference in counseling couples with a twin pregnancy, while studies on placental transfer of vaccine-derived antibodies are currently not available, especially in countries experiencing a vaccine shortage or unavailability of mRNA vaccines.

## Figures and Tables

**Table 1 vaccines-11-00116-t001:** Comparisons of immunization profiles between the mother and both twins.

	Mother	Twin A	Twin B
Anti-spike protein IgG levels (BAU/mL)	313.349	678.219	874.853
Neutralization activity			
% Inhibition (Wuhan)	70.453	86.764	88.088
% Inhibition (Delta)	56.632	75.745	76.645
Antibody subtypes			
IgG 1 *	0.289	0.631	0.623
IgG 2 *	0.012	0.021	0.029
IgG 3 *	0.007	0.002	0.000
IgG 4 *	0.011	0.014	0.003
IgA *	2.919	1.085	1.061
IgM *	0.406	0.003	0.005
Clinical data of the newborns			
Apgar scores at 5 min	-	9	9
Birth weight (g)	-	2825	2670
Sex	-	Male	Male
Structural anomaly	-	None	None
Hb (g/dl) / Hct (%)	-	13.7/42.3	13.9/43.0

* The data show the net OD values at 450 nm. OD = optical density.

## Data Availability

The datasets analyzed during the current study are available from the corresponding author upon reasonable request.

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
