# Peer review of "Placental Transfer Immunity to the Newborns in a Twin Pregnant Women Vaccinated with Heterologous CoronaVac-ChAdOx1"

_vaccines, 2023, doi:10.3390/vaccines11010116_

Round 1

Reviewer 1 Report

The aim of this study is to determine maternal SARS-CoV-2 antibody that can transfer to fetus through the placenta in mother who vaccinated with two doses of heterologous CoronaVac and AZD1222 vaccination

The introduction of the study, the results are clearly described and presented, and the discussion is appropriate. The abstract accurately describes the results and conclusions. The findings and conclusions, expand the knowledge of placenta transferred of vaccine-induced antibody against SARS-CoV-2 and the ability of virus neutralization against wild-type and delta variants.

Comments to authors: 

1.      In line 87-99, Could authors provide the threshold of anti-spike protein detection, which level defined as positive or negative and how to interpret the results of isotype of anti-spike antibody?

-  The details of mother and twin should be give be more information. I do not know which results belong to twin A or twin B, sex?? Dizygotic twin, the same sex or different sex, has any abnormality? or the authors can put these data in the the table. From our experience, twin who has more weight, has more antibody. Do those twin has hematocrit or other lab test?

2.      In line 100-104, Could author briefly provide more details about how to detect the neutralizing antibody. 

- The reagents should follow with Company, City and country, except the USA use the abbreviation of the state instead of USA.

- - The authors should discuss why the antibody levels in both twins were two times higher than those in the mothers who received CoronaVac/ChAdOx-1. As most of the literature found that the cord to maternal ratio of antibody specific to SARS-CoV-2 is around 1.0-1.2. This ratio of two times higher in cord blood compared to maternal blood is unusally high compared to the previous studies on SARS-CoV-2 and also other pathogens (such as measles and B.pertussis) The authors can use the ref of the other pathogens as a cited reference, eg pertussis, measles etc.

3.      In Table 1, please check the number after the decimal point.

4.      To help the reader, the authors should be added the abbreviation of gestational diabetes mellitus as GDM in line 77 due to the author mention this abbreviation in line 180.

Author Response

Point-by-point response

Reviewer 1

Comments and Suggestions for Authors

The aim of this study is to determine maternal SARS-CoV-2 antibody that can transfer to fetus through the placenta in mother who vaccinated with two doses of heterologous CoronaVac and AZD1222 vaccination

The introduction of the study, the results are clearly described and presented, and the discussion is appropriate. The abstract accurately describes the results and conclusions. The findings and conclusions, expand the knowledge of placenta transferred of vaccine-induced antibody against SARS-CoV-2 and the ability of virus neutralization against wild-type and delta variants.

Comments to authors:

  1. In line 87-99, Could authors provide the threshold of anti-spike protein detection, which level defined as positive or negative and how to interpret the results of isotype of anti-spike antibody?

Response: To define cut-off value, the WHO International Standard for anti-SARS-CoV-2 immunoglobulin (human) (NIBSC code: 20/136), which was assigned an arbitrary unitage of 1000 BAU/mL were used as positive control to make a standard curve. Antibody levels were determined from a standard curve and presented as BAU/mL. Pooled serum before COVID-19 pandemic was used as negative control. The cut-off value was assigned as ≥ 3 times of standard deviation of negative control value which was 58.5 BAU/mL.

The results of isotype of anti-spike antibody were presented as the net OD at 450 nm which were calculated by subtracting the OD value of wells coated with the antigen (recombinant SARS-CoV-2 Spike protein) by the OD value of antigen-free wells.

We have provided these explanations of cut-off value and isotype of anti-spike antibody value in the revised manuscript..

-  The details of mother and twin should be give be more information. I do not know which results belong to twin A or twin B, sex?? Dizygotic twin, the same sex or different sex, has any abnormality? or the authors can put these data in the the table. From our experience, twin who has more weight, has more antibody. Do those twin has hematocrit or other lab test?

Response: The more information of the newborns is added in Table 1. Placental chorionicity is added in “Case Presentation”, but zygosity test by DNA is not determined.

  1. In line 100-104, Could author briefly provide more details about how to detect the neutralizing antibody.

Response: We have added a brief detail of the neutralizing antibody detection procedure in par Laboratory.

- The reagents should follow with Company, City and country, except the USA use the abbreviation of the state instead of USA.

Response: The details of the companies of reagents used are added as suggested in part Laboratory.

- - The authors should discuss why the antibody levels in both twins were two times higher than those in the mothers who received CoronaVac/ChAdOx-1. As most of the literature found that the cord to maternal ratio of antibody specific to SARS-CoV-2 is around 1.0-1.2. This ratio of two times higher in cord blood compared to maternal blood is unusally high compared to the previous studies on SARS-CoV-2 and also other pathogens (such as measles and B. pertussis) The authors can use the ref of the other pathogens as a cited reference, eg pertussis, measles etc.

Response: The discussion on the levels are added as highlighted in the second and fifth paragraph of “Discussion”. (latency period is likely the cause of disparity)

  1. In Table 1, please check the number after the decimal point.

Response: The number of decimals are corrected to be homogeneous.

  1. To help the reader, the authors should be added the abbreviation of gestational diabetes mellitus as GDM in line 77 due to the author mention this abbreviation in line 180

Response: The abbreviation is added, as highlighted.

Reviewer 2 Report

Pongsatha, Chawansuntati, and Tongsong summarize the results of vaccinating a pregnant woman with heterologous Corona Vac-ChA-dOx1 and testing for the passive transfer of protective antibodies to the twin newborns.  This is a nice case study. My only concern is that in the discussion, very little attention is given to previous research of passive transfer.  Even if not much is known in the context of COVID-19 vaccination, there is some literature related to other vaccines and diseases that can be used for comparison.

In lines 116-136, a lot of comments are made without siting a source of that information.  Rather than saying there are very limited data about the types of vaccination regimens...  briefly summarize what is available in the literature.

In lines 133-134, reference where the information of teratogenic effect comes from

In lines 134-136, what data from the literature shows the benefits/reduced risks of vaccination for pregnant women?

in Table 1, what are the units of measure for IgG1 through IgM?

May want to include Prahl et al 2022 in your discussion (PMID 35908075)

Author Response

Point-by-point response

Reviewer 2

Comments and Suggestions for Authors

Pongsatha, Chawansuntati, and Tongsong summarize the results of vaccinating a pregnant woman with heterologous Corona Vac-ChA-dOx1 and testing for the passive transfer of protective antibodies to the twin newborns.  This is a nice case study. My only concern is that in the discussion, very little attention is given to previous research of passive transfer.  Even if not much is known in the context of COVID-19 vaccination, there is some literature related to other vaccines and diseases that can be used for comparison.

Response: In revised MS, discussion on other vaccinations with placental transfer to protect newborns are added, as highlighted in the second paragraph of “Discussion”

In lines 116-136, a lot of comments are made without siting a source of that information.  Rather than saying there are very limited data about the types of vaccination regimens...  briefly summarize what is available in the literature.

Response: In revised MS, the references / citations are provided, as highlighted in the first and second paragraph of “Discussion”

In lines 133-134, reference where the information of teratogenic effect comes from

Response: In revised MS, teratogenic effect is of concern from any medications during embryogenesis. However, no evidence has been well documented yet. We change the sentence to be “ .. is of theoretical concern”.

In lines 134-136, what data from the literature shows the benefits/reduced risks of vaccination for pregnant women?

Response: In revised MS, the references are provided as highlighted.

in Table 1, what are the units of measure for IgG1 through IgM?

Response: The values of IgG1 through IgM in Table 1 are the net OD (optical density) at 450 nm, calculated by the OD value of wells coated with SARS-CoV-2 Spike protein as antigen subtracted by the OD of antigen-free wells. Therefore, these values do not have units.

We have provided a detail of these values calculation in part laboratory and note in footnote of Table1.

May want to include Prahl et al 2022 in your discussion (PMID 35908075)

Response: In revised MS, the suggested study is now added and cited.
